# Molecular Mechanisms of Chitosan Interactions with Fungi and Plants

**DOI:** 10.3390/ijms20020332

**Published:** 2019-01-15

**Authors:** Federico Lopez-Moya, Marta Suarez-Fernandez, Luis Vicente Lopez-Llorca

**Affiliations:** Department of Marine Sciences and Applied Biology, Laboratory of Plant Pathology, Multidisciplinary Institute for Environmental Studies (MIES) Ramon Margalef, University of Alicante, 03080 Alicante, Spain; suarezfernandezmarta@gmail.com (M.S.-F.); lv.lopez@ua.es (L.V.L.-L.)

**Keywords:** chitosan, antimicrobial compounds, auxin, effectors, LysM motifs, plant immunity, ROS

## Abstract

Chitosan is a versatile compound with multiple biotechnological applications. This polymer inhibits clinically important human fungal pathogens under the same carbon and nitrogen status as in blood. Chitosan permeabilises their high-fluidity plasma membrane and increases production of intracellular oxygen species (ROS). Conversely, chitosan is compatible with mammalian cell lines as well as with biocontrol fungi (BCF). BCF resistant to chitosan have low-fluidity membranes and high glucan/chitin ratios in their cell walls. Recent studies illustrate molecular and physiological basis of chitosan-root interactions. Chitosan induces auxin accumulation in *Arabidopsis* roots. This polymer causes overexpression of tryptophan-dependent auxin biosynthesis pathway. It also blocks auxin translocation in roots. Chitosan is a plant defense modulator. Endophytes and fungal pathogens evade plant immunity converting chitin into chitosan. LysM effectors shield chitin and protect fungal cell walls from plant chitinases. These enzymes together with fungal chitin deacetylases, chitosanases and effectors play determinant roles during fungal colonization of plants. This review describes chitosan mode of action (cell and gene targets) in fungi and plants. This knowledge will help to develop chitosan for agrobiotechnological and medical applications.

## 1. Introduction

Chitosan is a linear polymer of beta-(1-4)-linked *N*-acetyl-2-amino-2-deoxy-d-glucose (acetylated) and 2-amino-2-deoxy-d-glucose (deacetylated [1]) subunits (Figure 1) [2]. Partial deacetylation of chitin by enzymatic or chemical processes generates chitosan [3]. Chitin is a key component of the cuticle of insects and exoskeleton of crustaceans, the cell wall of true fungi and that of some algae [1]. The main sources of chitosan production for commercial applications are marine crustaceans (mainly shrimps). The main way to synthetize chitosan is by chemical methods. Acids demineralize crustacean exoskeletons. Sodium hydroxide removes proteins from demineralized shells. Strong bases deacetylase chitin and yield chitosan [4]. Enzymatic methods can also produce chitosan. Chitin deacetylases (CDAs) generate chitosan from chitin. Combinations of chitin oligosaccharide deacetylases modify the pattern of polymer deacetylation [5,6,7]. Some fungal plant pathogens generate chitosan [8] during early infection to avoid host defenses [9]. CDA from the endophyte *Pestalotiopsis* sp. generates chitosan oligomers that unlike its chitin substrates no longer elicit plant immunity [10]. The nematophagous fungus *Pochonia chlamydosporia* produces chitosan during *Meloidogyne javanica* egg infection [11]. The fungus overexpresses CDA and chitosanase encoding genes in this process [11]. The activity of the catalytic center of a CDA encoded in the *P. chlamydosporia* genome has been recently confirmed experimentally [12].

Degree of deacetylation and molecular weight are crucial parameters for chitosan bioactivity. Chitosans usually have a degree of acetylation of less than 10% [3]. Numbers of *N*-acetyl glucosamine and glucosamine subunits in the molecule define chitosan molecular weight (M*w*) [3]. Large-Mid (70–100 kDa) M*w* chitosans are soluble in weak acid solutions (e.g., hydrochloric acid, citric acid, acetic acid). Only chitosan oligosaccharides (<5000 Da) are water-soluble. Protonation of chitosan amino groups generates positive charges. Both positive charge and amenability to structural modifications confer chitosan numerous biological properties. Wide varieties of industries use chitosan for different applications. Chitosan has potential for drug-delivery [3,13,14] or as a source of biomaterials to generate nanofibers or nanoparticles [15,16,17]. This review describes the mode of action of chitosan in fungi and plants. Figure 2 shows the main features summarized in this paper. Our main objectives are: (1) Analyze antimicrobial activity of chitosan; (2) Describe cell and gene targets for chitosan in fungi; (3) Illustrate agrobiotechnological applications of chitosan; (4) Discuss the effect of chitosan on plant growth and immunity.

## 2. Chitosan as Antimicrobial Agent

Chitosan is a versatile compound with antimicrobial activity [18,19,20,21]. Several studies have investigated the mode of action of chitosan [22,23,24]. In this review, we present an overview on the state-of-the-art of chitosan as a natural fungicide. Chitosan affects germination and hyphal morphology of economically important post-harvest fungal pathogens (e.g., *Rhizopus stolonifer* and *Botrytis cinerea*) [25,26,27,28]. This polymer also inhibits the growth of many other plant pathogenic and mycoparasitic fungi (such as *Alternaria* spp., *Colletotrichium* spp. or *Trichoderma* spp.) [20,29,30,31]. 

Chitosan has a great potential as antifungal agent to treat diseases caused by human pathogenic fungi [32,33,34,35,36]. Sensitive fungi show energy-dependent plasma membrane permeabilisation by chitosan [37]. This polymer also displays antibiotic activity against pathogenic bacteria [23,38,39,40,41]. Just as for fungi, chitosan permeabilises bacterial plasma membranes [36]. Recent studies suggest the use of chitosan as antimicrobial for clinical use [42]. This polymer kills opportunistic human pathogens such as *Fusarium proliferatum* and *Hamigera avellanea* [43]. Chitosan also arrests germination and growth of *Fusarium oxysporum* f.sp. *radicis lycopersici* and *Verticillium dahliae* [29]. These fungi are phylogenetically close to opportunistic human pathogens. This polymer also restricts growth of other fungal human pathogens such as *Aspergillus fumigatus* and *Rhizopus stolonifer*. Chitosan inhibits biofilm formation in *A. fumigatus* [44]. Limitation of nutrients (carbon; C and nitrogen; N) enhances chitosan antifungal activity to human pathogens [43]. Deprivation of nutrients modifies cell wall architecture which affects fungal growth [43,45,46,47]. To this respect, low branching (glucan content) of fungal cell wall increases sensitivity to chitosan (Figure 2) [11]. There is a direct link between cell wall and membrane since the synthesis of key cell wall components (glucans and chitin) is performed by plasma membrane-associated synthase complexes [48,49]. Chitosan inhibits growth of *Candida* spp. and *Cryptococcus* spp. pathogenic yeasts [34,36,42,50,51,52]. As for filamentous fungi, chitosan inhibits *C. albicans* growth under the same nutritional status (carbon) as in human blood (glycemia). Chitosan significantly reduces *C. albicans* virulence on *Galleria mellonella* L. under these conditions [36]. *G. mellonella* is a well-established model host to test the effect of antimicrobials on virulence of fungal and bacterial pathogens [53,54,55]. This polymer is harmless to mammalian (human and monkey) cells at concentrations fungicidal to human fungal pathogens [43]. This makes it suitable for clinical application. Nanotechnology allows new chitosan formulations (e.g., nanofibres, nanocomposites, or nanocapsules) with potential for drug delivery, in medical mycology and biotechnology [24,56,57,58,59,60,61]. These findings open-up new possibilities to develop chitosan as a natural antifungal compound for practical use.

## 3. Chitosan Alters Gene Expression in Fungi

Several studies have investigated the role of plasma membrane in the sensitivity of fungi to chitosan [37,62]. The membranes of chitosan-sensitive fungi (e.g., *Neurospora crassa*) are highly-fluid (rich in polyunsaturated free fatty acid (FFA) such as linolenic acid). On the contrary, chitosan-resistant fungi (*P. chlamydosporia*) have low-fluidity membranes (enriched on saturated FFA such as palmitic or stearic acid [62]). Recently, our group found that chitosan permeabilises *N. crassa* plasma membrane using flow cytometry. This triggers intracellular production of reactive oxygen species (ROS) and cell death [43]. RNAseq data and gene ontology (GO) analysis reveals oxidoreductase activity, plasma membrane and transport as main categories induced by chitosan [36]. Chitosan also enriches oxidative metabolism, respiration and transport GO functions in the model yeast *Saccharomyces cerevisiae* [63]. *S. cerevisiae* plasma membrane, response to stress and cell wall integrity genes are induced by chitosan [64]. Testing deletion strain mutants confirms Lipase Class III, Monosaccharide transporter and Glutathione transferase encoding genes (NCU03639; NCU04537; NCU10521, respectively) as main chitosan targets in *N. crassa* [36]. They might play a role in membrane repair, assimilation of catabolites and buffering ROS surplus derived from chitosan damage. Antifungal proteins, such as PAF from *Penicillium chrysogenum*, have a mode of action similar to chitosan since they also permeabilise plasma membranes and induce ROS production [65,66]. Synthesis of oxidative by-products of metabolism reflects (in peroxisomes and mitochondria) the energetic status of the cell. This would explain why plasma membrane permeabilisation by chitosan is an energy dependent process [37]. A chemical or physiological block of the electron transport chain abolishes the antifungal activity of chitosan in *N. crassa* [37]. Peroxisome and mainly mitochondria, the main organelles involved in ROS generation, support the relevance of ROS metabolism in the response of *N. crassa* to chitosan. The early response of *N. crassa* to chitosan involves partial membrane permeabilisation and the onset of ROS production [43]. We hypothesize that chitosan causes an intracellular ROS burst which starts oxidizing FFA from cell membranes. Increased membrane oxidation finally leads to full plasma membrane permeabilisation and is perhaps responsible for the antifungal effect of chitosan. Induction of ROS and growth inhibition also happen under glucose starvation in *Candida glabrata* [67,68]. This again links nutrient content, ROS and antifungal action and may explain results described above with chitosan. Future studies should investigate the generation of oxylipins, metabolites derived from lipid peroxidation, [69] by chitosan and its relationship with the cellular nutritional status. Recent works describe the involvement of Ca^2+^ on plasma membrane remodeling during cell fusion in *N. crassa* [70,71] and *S. cerevisiae* [72]. This would explain why Ca^2+^ increases tolerance to chitosan in *N. crassa* [36]. On the contrary, *N. crassa syt1* deletion strains involved in plasma membrane homeostasis mediated by Ca^2+^ [73,74] are more sensitive to chitosan than the wild-type strain [36]. Figure 3 shows a model we propose on the mode of action of chitosan on sensitive fungi. The link of ROS, membrane homeostasis, Ca^2+^ and chitosan is therefore an attractive subject for future studies [75,76,77]. 

## 4. Chitosan Acts as Gene Modulator in Tolerant Fungi

Chitosan can be combined with tolerant fungi such as biocontrol fungi (BCF) [29,57,78,79]. BCF can degrade chitosan using it as a nutrient source [29,80]. Nematophagous fungi such as *P. chlamydosporia* can withstand high doses of chitosan. The genome of *P. chlamydosporia* [81] reveals an expansion of hydrolases. This may reflect the multitrophic (saprotrophic, endophytic, nematophagous) behavior of the fungus. *P. chlamydosporia* encodes enzymes to generate and degrade chitosan such as chitin deacetylases or chitosanases [81]. Recent studies show that chitosan induces these genes in *P. chlamydosporia* (mainly *cda1*, *csn1* and *csn6*) during nematode egg infection [12]. Other studies demonstrate that chitosan also induces *P. chlamydosporia* proteases [79]. Proteomics reveals that chitosan alone induces expression of *vcp1* serine protease also involved in nematode egg infection by *P. chlamydosporia* [82]. Chitosan also induces accumulation of *vcp1* and *scp1* (a serine carboxypeptidase) proteases in appressoria of *P. chlamydosporia* infecting root-knot nematode eggs, enhancing virulence [79]. Chitosan also increases sporulation of BCF (*P. chlamydosporia* and *Beauveria bassiana*; [29]). Other studies reveal that chitosan enhances growth and sporulation in mycoparasitic biocontrol fungi *Trichoderma* spp. (such as *T. koningiopsis*) [31]. However, other *Trichoderma* spp. are hypersensitive to chitosan such as *T. harzianun* or *T. neocrassum* [29,31]. Tolerance of fungi to chitosan depends on the FFA composition as we have explained above. *T. knoningiopsis,* resistant to chitosan shows plasma membrane enriched in saturated FFA. Contrarily, other *Trichoderma* spp. highly sensitive to chitosan display a large content of poly-unsaturated FFAs [31]. The use of chitosan in combination with BCF opens new environmentally friendly possibilities to manage pest and diseases caused by insects, nematodes or fungi.

## 5. Agrobiotechnological Applications of Chitosan

### 5.1. Chitosan in Plant Protection

Abuse of pesticides and fertilizers generates environmental pollution, eutrophication and loss of biodiversity [83,84,85]. Chitosan is widely used in agriculture on pre- and postharvest treatments of crops to control microbial infections [56,86,87]. Chitosan can solve these problems minimizing pollution. For example, nanoparticles of chitosan and porous silica can encapsulate pesticides and fertilizers and reduce their environmental impact [88]. Application of chitosan to plants protects them against infections caused by important plant pathogenic fungi such as *B. cinerea* or *F. oxysporum* f.sp. radicis-lycopersici [89,90,91]. Chitosan promotes parasitism of root-knot nematode eggs by *P. chlamydosporia* [79]. In combination with *Hirsutella*, chitosan can control cyst nematodes in soybean [92]. In recent studies, chitosan enhances pathogenicity of *P. chlamydosporia* to root-knot nematode (RKN) eggs in greenhouse and semi-field experiments [93]. This opens up new strategies for sustainable management of plant-parasitic nematodes in economically important crops such as tomato, barley or soybean. 

### 5.2. Chitosan in Crop Growth and Defenses

This polymer also stimulates seed germination and growth of seedlings of ornamental plants [94,95]. Chitosan is also an elicitor of plant defenses in economically important crops [96,97]. Effects of chitosan on plant development and immunity are revised below. 

## 6. Chitosan Effect on Plant Development

Most chitosan treatments to plants have been performed on leaves. High doses of chitosan in the rhizosphere of *Arabidopsis*, tomato or barley significantly arrest root development [98]. Chitosan alters root cell morphology and pattern of division. This abolishes polarity of root apex growth, stops elongation and modifies root architecture. *Arabidopsis thaliana* is an adequate tool to analyze the mechanisms of root growth inhibition by chitosan. Chitosan causes accumulation of auxin (mainly indole acetic acid; IAA) in *Arabidopsis* roots. This accumulation is mediated by induction of Tryptophan-dependent biosynthesis pathway genes (*yuc2, aao1, ami1*) and by repression of the main gene (*pin1*) involved in IAA translocation [98] (Figure 4). Auxin build-up reduces primary root length and alters secondary root emergence [99]. Chitosan also affects regulation of gibberellic acid [100]. This polymer also alters patterns of expression of stress-related hormones jasmonic (JA) and salicylic acid (SA) genes in roots [98]. These transcriptomical and physiological changes can be elicited by other abiotic stresses [101,102]. Stress caused by application of high chitosan doses causes JA, SA and ROS accumulation in *Arabidopsis* roots together with other pleiotropic effects [98]. Cross-talk between abiotic and biotic stress responses in plants involves ROS and hormone signaling [103]. Therefore, chitosan irrigation treatments must avoid overstress to plant roots. Chitosan concentrations used in agricultural practices should restrict growth of pathogens, but favor growth of beneficial fungi (BCF) and plants. 

## 7. Plant Immunity: Role of Chitin/Chitosan

Chitin is a key structural component of fungal cell walls. Plant chitinases and glucanases release chitin and β-glucan oligomers from cell walls of fungal pathogens. These oligomers (MAMPs, microbe-associated molecular patterns) are recognised by plant membrane PRRs (pattern recognition receptors) triggering a PTI (pattern-triggered immunity [104]) response. Some fungal pathogens can modify their cell walls during plant tissue colonisation, turning chitin into chitosan [8,105]. This may protect hyphae of fungal pathogens from plant chitinases. Very low concentrations (picomolar; pM) of chitin oligomers can still trigger PTI [106]. Chitin oligomers with large degrees of acetylation and polymerisation induce main PTI responses (ROS production and alkalinisation) [107,108,109]. Chitosan induces callose deposition in plants [110]. Plant immunity also involves overexpression of PR (Pathogenesis-Related) proteins in response to fungi and other pathogens. PR-4 family includes class-I and -II chitinases, responsible for chitin degradation [111,112]. Chitosan induces expression of PR proteins (NPR1) in roots [98] and leaves (PR1 and PR5) [113]. Plant chitinases show less affinity for chitosan than chitin. Therefore, chitosan is a less efficient MAMP than chitin [8,108]. Chitosan is also a component of the cell wall in germ tube and appressoria in the fungal pathogen *Magnaporthe oryzae* [114]. *M. oryzae* expresses CDAs during appressorium development. Deletion of these genes compromises conidia germination, adhesion and appressoria differentiation. Application of exogenous chitosan restores normal development [114]. *M. oryzae* chitosan on the surface of the leaf gives stimuli to promote adhesion, appressorium development and plant infection. Therefore, chitosan could promote evasion of plant immunity and fitness/virulence of fungal pathogens, including biocontrol agents like *P. chlamydosporia*. Plant pathogenic fungi also induce other structural changes in their cell walls to avoid plant defenses such as accumulation of α-1,3-glucan. Masking chitin in fungal cell walls with α-1,3-glucan also provides protection from plant chitinases. This delays PTI activation triggered by MAMP (chitin oligomers) recognition [8]. 

Fungal pathogens also avoid plant immunity shielding cell wall chitin with effectors. These mostly secreted proteins facilitate pathogen entry into the host and modulate plant immune perception (PTI) altering host physiology for pathogen’s benefit [115]. LysM effectors bind chitin and chitin oligomers limiting PTI responses [116]. In this way, they protect fungal pathogens from plant chitinases [8]. They also prevent recognition of chitin oligomers (MAMPs) by plant PRR. Chitin elicitor binding proteins (CEBiP/CERK) are essential in chitin perception by plants. However, chitosan does not have specific receptors in plants [117]. This would be an additional reason why chitosan does not efficiently activate PTI responses. LysM effectors could protect fungal plant pathogens avoiding perception by mycoparasitic fungi [118]. Likewise, LysM domains can recognise peptidoglycan and could agglutinate bacterial competitors of plant pathogenic fungi. Just like pathogenic fungi, mutualistic endophytes (including mycorrhizae) also secrete LysM effectors to suppress plant immunity. These fungi can manipulate plant hormone signaling associated with plant defenses [115,119]. Chitin derivatives (Nod and Myc factors) act as symbiotic signaling compounds for rhizobia and mycorrhiza [120,121]. LysM proteins protect entomopathogenic fungi from insect immune system [122]. *B. bassiana* activates *Blys2* and *Blys5* (containing five and two LysM domains, respectively) expression during insect infection. *Blys5* besides chitin can also bind chitosan and cellulose. Both LysM proteins are essential for full fungal virulence and could protect hyphae from host chitinases. 

## 8. Concluding Remarks

Chitosan is a versatile compound with a wide range of applications. In the clinical field, chitosan can inhibit filamentous fungi (e.g., *Aspergillus fumigatus*) and yeast (e.g., *Candida albicans*) human pathogens. Application of chitosan under low nutrient (C and N) status favors the antifungal mode of action of this polymer on human pathogenic fungi. Chitosan also reduces development of the model fungus *N. crassa.* Chitosan drastically reduces *N. crassa* spore germination and growth development. *N. crassa* spores exposed to chitosan induce expression of a lipase Class III, a monosaccharide transporter and a glutathione transferase encoding genes. These are the main chitosan targets in this filamentous fungus. A membrane protein ARL1 is the main chitosan target in the model yeast *S. cerevisiae* [63]. These proteins could be chitosan targets in fungi and their study could help to develop chitosan as a drug. On the other hand, chitosan is compatible with several BCF (*P. chlamydosporia, B. bassiana* or *Trichoderma* spp.). A high amount of saturated FFA in the plasma membrane together with a high glucan/chitin ratio in the cell wall make fungi resistant to chitosan (Figure 2). Chitosan is also compatible with biocontrol fungi applied *in planta*. However, application of chitosan on plant should be highly regulated. High doses of chitosan could block plant (tomato and barley) growth and development affecting root meristem architecture and organization. In *Arabidopsis,* the main transcription factor (WOX5) that regulates cell division in the root quiescent center is repressed by chitosan [98]. Besides, chitosan also alters hormone homeostasis. Chitosan stimulates auxin (IAA) accumulation in roots avoiding its translocation. Besides auxin, JA and SA accumulation in roots would explain changes of cell organisation in *Arabidopsis* roots exposed to chitosan. Chitosan and chitin also play a key role controlling plant immunity. Mainly chitin but also chitosan act as MAMPs triggering a PTI immune response in plants. Chitosan could play a crucial role for fungal endophytes and pathogens colonizing plants generating a fight between plant immunity and fungal virulence.

## Figures and Tables

**Figure 1 ijms-20-00332-f001:**
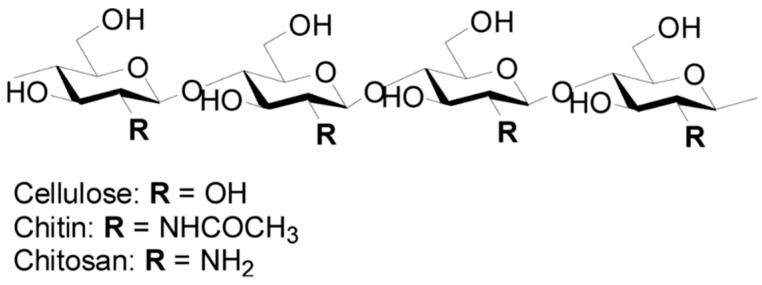
Chitosan, chitin and cellulose molecular structures (Modified from [2]).

**Figure 2 ijms-20-00332-f002:**
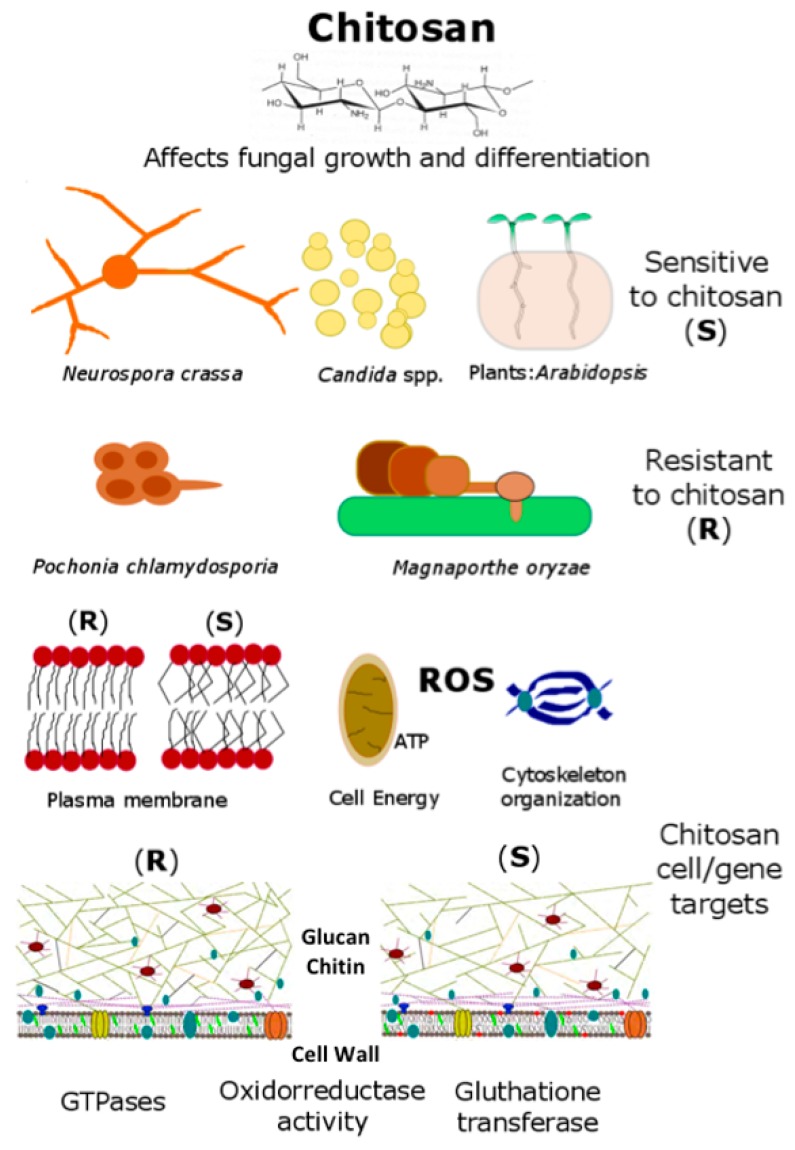
Conceptual diagram of chitosan as antifungal and gene modulator in fungi and plants (modified from [18]).

**Figure 3 ijms-20-00332-f003:**
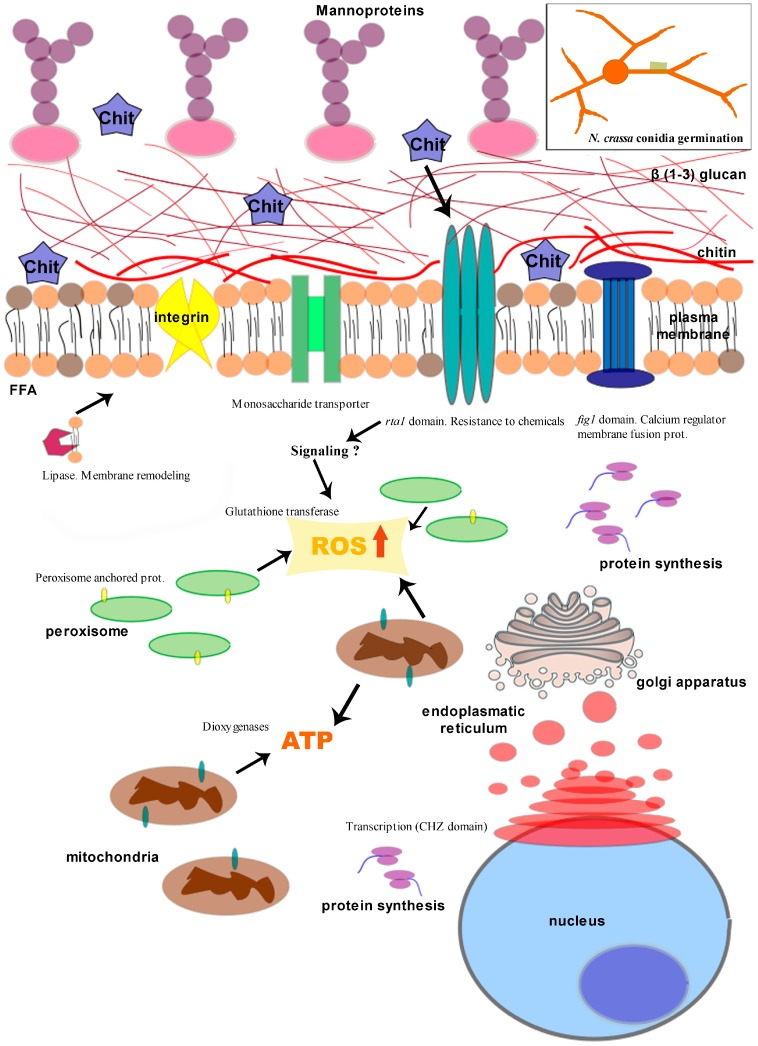
Proposed mode of action of chitosan as a gene modulator on sensitive fungi. A lipase modifies the plasma membrane increasing its permeability and Ca^2+^ would mediate this in association with a calcium regulator membrane fusion protein with a FIG1 domain. Moreover, a monosaccharide transporter could be involved in the assimilation or detoxification of monomers of *N*-acetyl glucosamine. Besides, a glutathione transferase and two dioxygenases may be associated with the response of the fungus to oxidative stress caused by chitosan increasing reactive oxygen species (ROS) and ATP production. Finally, the mechanisms of protein synthesis (peroxisome-anchored protein) and resistance to chemicals (RTA1 domain) have also modifications in their gene expression when exposed to chitosan (modified from [36]).

**Figure 4 ijms-20-00332-f004:**
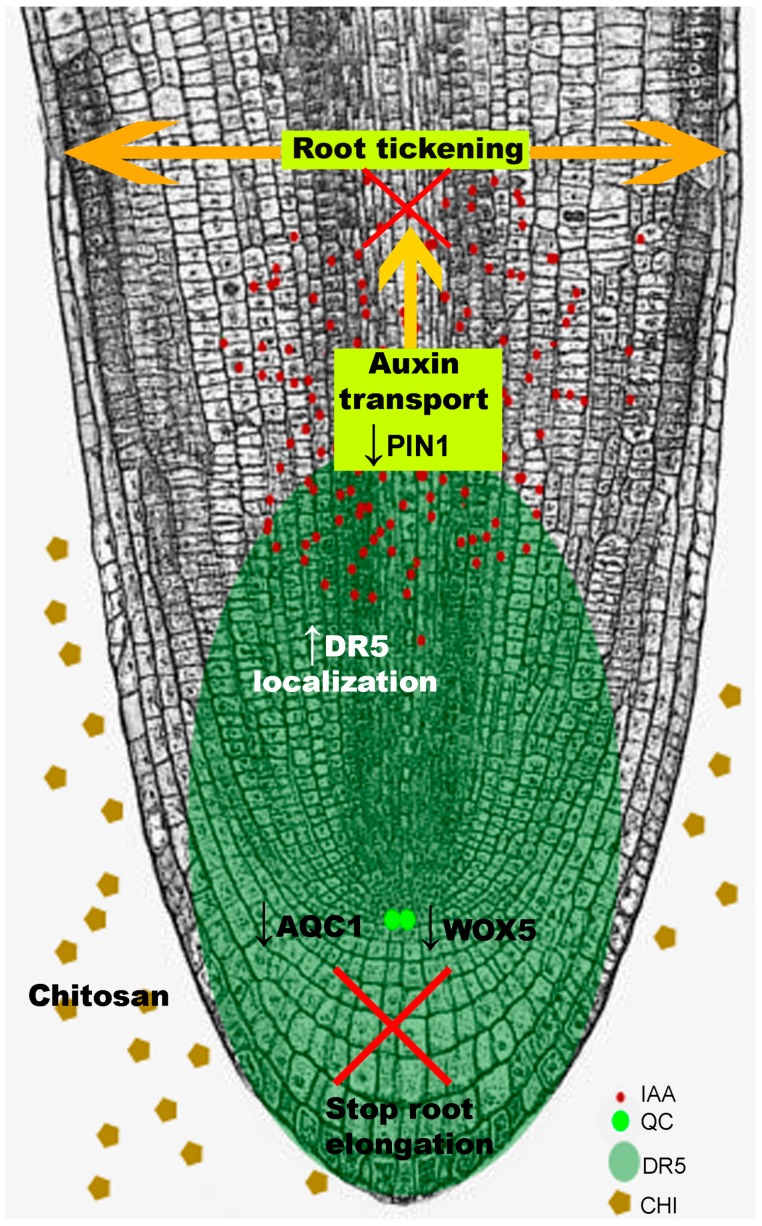
Chitosan causes auxin accumulation in roots. This is shown by lack of auxin transport in the vascular system (no DR5 expression). This auxin accumulation represses *wox5*, which controls activity of the quiescent center and in turn root polar growth. As a result, root stops elongating and root apex thickens. ROS and phenolic accumulation are associated with programmed cell death and stress-hormone (JA, SA) accumulation.

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
