# Peer review of "Molecular Mechanisms of Chitosan Interactions with Fungi and Plants"

_ijms, 2019, doi:10.3390/ijms20020332_

Reviewer 1 Report

The manuscript reviews the mode of antimicrobial action and molecular mechanisms of interaction of chitosan with bacteria and fungi as well as its application as a biocontrol agent against plant pathogenic fungi. The general impression from reading the manuscript is that the manuscript is clearly written, and figures are clearly

submitted. The manuscript reviews most of the recent results published on the subjects so far.

Conclusion: the manuscript should be accepted.

Author Response

The manuscript reviews the mode of antimicrobial action and molecular mechanisms of interaction of chitosan with bacteria and fungi as well as its application as a biocontrol agent against plant pathogenic fungi. The general impression from reading the manuscript is that the manuscript is clearly written, and figures are clearly submitted. The manuscript reviews most of the recent results published on the subjects so far.Conclusion: the manuscript should be accepted.

Thank you.

Reviewer 2 Report

Molecular mechanisms of growth and development inhibition in fungi and plants by chitosan

By Lopez-Moya et al

It is a very interesting paper, that provides a summary of new an important knowledge about chitosan and the effects in the development and growth of fungi and plants. However, I have several comments that I would like to request before the paper can be published.

I strongly recommend to review the structure of the abstract. It should have first an introduction, then highlight the importance of the paper (what makes it unique) and then the scope and the objective of the work. Right now, it has good information, but does not look organized. Also, I recommend to check the grammar and English edition.

The keywords should be organized in alphabetic order.

Introduction

Figure 1 should be improved; the letters should have the same size than the rest of the text.

It is not really clear the purpose of the review in the introduction. It becomes clear after section 2. I recommend to clarify the introduction for this purpose. The objectives should be presented in the introduction and not in the next sections of the paper.

Again, I recommend to make an extensive English editing, since chitosan is mentioned almost in every sentence as the first word and becomes tiring the lecture.

The subtitle in section 3, should be more compact like “Chitosan effects in fungi”; avoid the description in the title “Chitosan alters sensitive fungi…” it looks more like a description than a title.

Line 96: have low-fluidity membranes (enriched on saturated FFA such as palmitic or stearic acid; [60]) remove the semicolon.

Even if the figure 2 is interesting, it needs a full description of what is happening there and what is described. I cannot see the description of the figure which could be very useful to understand the process of the antifungal activity of chitosan and the relationship with ROS mechanisms.

Line 161: “Application of chitosan to plants protects them…” it should be: “Application of chitosan to plants protect them…”

In my opinion, section 5 could be improved, adding more information, with a table that includes example of applications and references, since this will increase the importance of the next section. Even, section 5 could be divided in sub-sections for the different agricultural applications. This is a very important and promising field for chitosan.

Between lines 179-180, it should be very interesting to describe how do López-Moya et al. studied the mechanism of root growth inhibition by chitosan. Also, figure 3 needs more explanation and description to understand it better. At the end of this section, it is important to highlight what are the main consequences of the accumulation of JA, SA and ROS.

Line 203 “Defence” should be “Defense”.

Figure 4 should be described in the text not in the concluding remarks a section that is better to summarize the main aspects of the paper and the perspectives after the literature review.

It should be added in the figure 4 pictures of the composition of the cell wall that makes fungi sensitive or not to chitosan to summarize better the concept.

I recommend to add a section of the fungi described here and the main aspects of pathogeneicity or economic importance and also, their chemical composition.

Some of the statements (for example line 72: “Chitosan restricts growth of other fungal human pathogens such as Aspergillus fumigatus and Rhizopus stolonifer . Chitosan mainly acts as inhibitor of biofilm formation in A. fumigatus [42]. Just as for the model fungus Neurospora crassa , limitation of nutrients (C and N) chitosan also enhances antifungal activity to human pathogens [41].”) should include plots as a result of measurements and analysis of experiments published in the original papers that should be mentioned and explained. It is important that people can understand that statement comes from analysis of ex

Author Response

It is a very interesting paper, that provides a summary of new an important knowledge about chitosan and the effects in the development and growth of fungi and plants. However, I have several comments that I would like to request before the paper can be published.

I strongly recommend to review the structure of the abstract. It should have first an introduction, then highlight the importance of the paper (what makes it unique) and then the scope and the objective of the work. Right now, it has good information, but does not look organized. Also, I recommend to check the grammar and English edition.

We have changed the structure of the abstract as recommended.

The keywords should be organized in alphabetic order.

Corrected.

Introduction

Figure 1 should be improved; the letters should have the same size than the rest of the text.

We have modified the size of the figure so letters now have the same size than the rest of the text.

It is not really clear the purpose of the review in the introduction. It becomes clear after section 2. I recommend to clarify the introduction for this purpose. The objectives should be presented in the introduction and not in the next sections of the paper.

We have added Figure 4 (now Figure 2) to the introduction. We quote this figure to explain the purposes of the paper and present our main objectives in the review.

Again, I recommend to make an extensive English editing, since chitosan is mentioned almost in every sentence as the first word and becomes tiring the lecture.

We tried to correct this by avoiding repeating ‘chitosan’ so much.

The subtitle in section 3, should be more compact like “Chitosan effects in fungi”; avoid the description in the title “Chitosan alters sensitive fungi…” it looks more like a description than a title.

The subtitle in section 3 has been changed to “Chitosan effects in fungi”.

Line 96: have low-fluidity membranes (enriched on saturated FFA such as palmitic or stearic acid; [60]) remove the semicolon.

Changed.

Even if the figure 2 is interesting, it needs a full description of what is happening there and what is described. I cannot see the description of the figure which could be very useful to understand the process of the antifungal activity of chitosan and the relationship with ROS mechanisms.

Description of processes occurred in figure 2 (now, figure 4) has been added to the figure caption.

Line 161: “Application of chitosan to plants protects them…” it should be: “Application of chitosan to plants protect them…”

Changed.

In my opinion, section 5 could be improved, adding more information, with a table that includes example of applications and references, since this will increase the importance of the next section. Even, section 5 could be divided in sub-sections for the different agricultural applications. This is a very important and promising field for chitosan.

We have rearranged the text so that applications of chitosan in plant protection, development-immunity are two new subsections.

Between lines 179-180, it should be very interesting to describe how do López-Moya et al. studied the mechanism of root growth inhibition by chitosan. Also, figure 3 needs more explanation and description to understand it better. At the end of this section, it is important to highlight what are the main consequences of the accumulation of JA, SA and ROS.

We have added the requested information to the figure legend

Line 203 “Defence” should be “Defense”.

Changed.

Figure 4 should be described in the text not in the concluding remarks a section that is better to summarize the main aspects of the paper and the perspectives after the literature review.

It should be added in the figure 4 pictures of the composition of the cell wall that makes fungi sensitive or not to chitosan to summarize better the concept.

Composition of the structure of the cell wall was added to Figure 4, and it is now Figure 2  (introduction)

I recommend to add a section of the fungi described here and the main aspects of pathogeneicity or economic importance and also, their chemical composition.

Detailed information on plasma membrane and cell wall composition in fungi and plants is not widely available.

We provide that mainly for fungi, which we have initially evaluated their sensitivity to chitosan. Similar studies for plants are in progress now in our laboratory.

Some of the statements (for example line 72: “Chitosan restricts growth of other fungal human pathogens such as Aspergillus fumigatus and Rhizopus stolonifer . Chitosan mainly acts as inhibitor of biofilm formation in A. fumigatus [42]. Just as for the model fungus Neurospora crassa , limitation of nutrients (C and N) chitosan also enhances antifungal activity to human pathogens [41].”) should include plots as a result of measurements and analysis of experiments published in the original papers that should be mentioned and explained. It is important that people can understand that statement comes from analysis of ex

We have added a new figure (figure 3) for a better understanding of the readership.

Reviewer 3 Report

This manuscript is a review article written by focusing on the function of chitosan: antifungal activity, and biological effects on fungi and plants. First impression of this manuscript by reading thoroughly was that the contents of this paper is too much, and it’s so tough to read out from the beginning to end. Honestly I could not realize the meanings of the title and the last sentence of abstract (what the authors would like to say in this paper?). I felt that this review is very descriptive and not emphasizing the points to read.

              I can understand the situation that the authors would like to publish a review paper with broad range of research topics because the corresponding and first author, Federico Lopez-Moya, have publications regarding the function of chitosan in various research area, from fungus to plants. However, I think that the most important point for authors to consider for writing the review paper is to get great interest from the readers of this journal, but not to forcibly put too much topics into one review paper. Therefore, I strongly recommend the authors to rewrite this manuscript only with appropriately chosen topics: omit the contents of “effects of chitosan on plant development” and long introduction (former part) of “Plant immunity: role of chitin/chitosan” because these seem to make confusion and complexity in this manuscript. Alternatively, it’s acceptable that the authors will take only the contents regarding chitosan function on plant except for long introduction (former part) of “Plant immunity: role of chitin/chitosan”. If the authors will totally revise the manuscript as my above recommendations and/or according to the authors’ idea by thinking about readers, I would like to ,of course, be willing to consider for publication in this journal.

Minor comments:

Caption of Figure 1, 2, 4: I could not find any references for “Ifuku, 2014” and “Lopez-Moya and Lopez-Llorca, 2016” in reference list. In addition, reference number should be noted (Figure 2).

Table 1: The explanation of PR proteins seems not be required because only PR-4 came out in this paper. Just refer a paper for it.

Reference: There are a lot of obvious errors and wrong styles (e.g. not italic, how to abbreviate journal names, underlined doi, not uniformed styles of references.…). Please reconfirm the style of all references prior to next submission.

Author Response

This manuscript is a review article written by focusing on the function of chitosan: antifungal activity, and biological effects on fungi and plants. First impression of this manuscript by reading thoroughly was that the contents of this paper is too much, and it’s so tough to read out from the beginning to end. Honestly I could not realize the meanings of the title and the last sentence of abstract (what the authors would like to say in this paper?). I felt that this review is very descriptive and not emphasizing the points to read.

Abstract and title have been rewritten accordingly.

I can understand the situation that the authors would like to publish a review paper with broad range of research topics because the corresponding and first author, Federico Lopez-Moya, have publications regarding the function of chitosan in various research area, from fungus to plants. However, I think that the most important point for authors to consider for writing the review paper is to get great interest from the readers of this journal, but not to forcibly put too much topics into one review paper. Therefore, I strongly recommend the authors to rewrite this manuscript only with appropriately chosen topics: omit the contents of “effects of chitosan on plant development” and long introduction (former part) of “Plant immunity: role of chitin/chitosan” because these seem to make confusion and complexity in this manuscript. Alternatively, it’s acceptable that the authors will take only the contents regarding chitosan function on plant except for long introduction (former part) of “Plant immunity: role of chitin/chitosan”.

The former part of “Plant immunity: role of chitin/chitosan” has been summarized, that part now shows mostly chitin/chitosan effects on plant immunity.

If the authors will totally revise the manuscript as my above recommendations and/or according to the authors’ idea by thinking about readers, I would like to, of course, be willing to consider for publication in this journal.

Minor comments:

Caption of Figure 1, 2, 4: I could not find any references for “Ifuku, 2014” and “Lopez-Moya and Lopez-Llorca, 2016” in reference list. In addition, reference number should be noted (Figure 2).

Those references have been added to bibliography.

Table 1: The explanation of PR proteins seems not be required because only PR-4 came out in this paper. Just refer a paper for it.

Table 1 was omitted and also most of the part of the text.

Reference: There are a lot of obvious errors and wrong styles (e.g. not italic, how to abbreviate journal names, underlined doi, not uniformed styles of references.…). Please reconfirm the style of all references prior to next submission.

We have made bibliography again in order to avoid mistakes.

Round  2

Reviewer 3 Report

I have carefully read the revised manuscript and the response to the reviewers' comments from authors. I got that the authors tried to refine their manuscript according to my and other reviewer’s comments, but I really regret that the revised manuscript is obviously incomplete as stated below. I think that the authors should submit the manuscript REVISED as far as possible (perfect one). They should check thoroughly prior to submission. It’s a general manner for submission of paper. Therefore, I could not unfortunately recommend this paper for publication even if the contents of this paper is so interesting to the researchers working on the physiological and biological roles of chitosan and related materials.

> Figure 3 is modified data taken from published paper. Should this figure be shown in this review? I could not get the necessity of this figure and think it should be just referred in text.

> As other reviewer pointed, I also recommend to make an extensive English editing by a native speaker because the text in this manuscript is pretty hard for me to read out. Some phrases are incomprehensible. For example, What is the meaning of “Describe chitosan cell and gene target in fungi (L56)?

> The authors said that “We have made bibliography again in order to avoid mistakes.” However, I can still find too much errors in reference list as I pointed previously.

> The list of keywords are suitable for this paper? Keywords should be noun. “Antifungal” is not noun.

> There are mismatches between reference numbers in text and the contents of referred papers (e.g. in the caption of Fig3 and Fig. 4).

Author Response

Dear Reviewer 3,

our response to your comments is in the enclosed file.

With best wishes,

Federico Lopez-Moya

Answers to the comments.

Reviewer #3:

I have carefully read the revised manuscript and the response to the reviewers' comments from authors. I got that the authors tried to refine their manuscript according to my and other reviewer’s comments, but I really regret that the revised manuscript is obviously incomplete as stated below. I think that the authors should submit the manuscript REVISED as far as possible (perfect one). They should check thoroughly prior to submission. It’s a general manner for submission of paper. Therefore, I could not unfortunately recommend this paper for publication even if the contents of this paper is so interesting to the researchers working on the physiological and biological roles of chitosan and related materials.

 We have followed your advice and have revised thoroughly the MS

> Figure 3 is modified data taken from published paper. Should this figure be shown in this review? I could not get the necessity of this figure and think it should be just referred in text.

Figure 3 has been removed from the MS and the information has been included in the text.

> As other reviewer pointed, I also recommend to make an extensive English editing by a native speaker because the text in this manuscript is pretty hard for me to read out. Some phrases are incomprehensible. For example, What is the meaning of “Describe chitosan cell and gene target in fungi (L56)?

We have modified the sentence in L.64. Now reads “Describe cell and gene targets for chitosan in fungi”. We have also carried out full English editing of the MS.

> The authors said that “We have made bibliography again in order to avoid mistakes.” However, I can still find too much errors in reference list as I pointed previously.

We have revised the bibliography and corrected mistakes in journal abbreviation and page numbers. DOI numbers have been removed.

> The list of keywords are suitable for this paper? Keywords should be noun. “Antifungal” is not noun.

Keywords have been also revised and modified: “Chitosan, Antimicrobial compounds, auxin, effectors, LysM motifs, plant immunity, ROS.”

> There are mismatches between reference numbers in text and the contents of referred papers (e.g. in the caption of Fig3 and Fig. 4).

 This has been corrected

Round  3

Reviewer 3 Report

I have carefully read the response to my comments from authors. Because the authors completely responded to my previous reviewing comments, I have no additional questions and comments on their revised manuscript.